# Correlation between Spectral Characteristics and Physicochemical Parameters of Soda-Saline Soils in Different States

**Xiaojie Li [1,\*], Jianhua Ren [2], Kai Zhao [1] and Zhengwei Liang [1,\*]**

[1] Northeast Institute of Geography and Agroecology, Chinese Academy of Sciences, No.4888 Shengbei Street, Gaoxinbei District, Changchun 130102, China; zhaokai@iga.ac.cn

[2] Harbin Normal University, No.1 Shida Road, Limin District, Harbin 150025, China; mariahdion@163.com

\* Correspondence: lixiaojie@iga.ac.cn (X.L.); liangzhengwei@iga.ac.cn (Z.L.); Tel.: +86-431-85542224 (X.L.)

**Abstract:** The spectral features of soils are a comprehensive representation of their physicochemical parameters, surface states, and internal structures. To date, spectral measurements have been mostly performed for powdered soils and smooth aggregate soils, but rarely for cracked soils; a common state of soda-saline soils. In this study, we measured the spectral features of 57 saline soil samples in powdered, aggregate, and cracked states for comparison. We then explored in depth the factors governing soil spectral features to build up simple and multiple linear regression models between the spectral features and physicochemical parameters (salt content, Na$^+$, pH, and electronic conductivity (EC)) of saline soils in different states. We randomly selected 40 samples to construct the models, and used the remaining 17 samples for validation. Our results indicated that the regression models worked more effectively in predicting physicochemical parameters for cracked soils than for other soils. Subsequently, the crack ratio (CR) was introduced into the regression models to modify the spectra of soils in powdered and aggregate states. The accuracy of prediction was improved, evidenced by a 2–11% decrease in the parameters mean absolute error (MAE).

**Keywords:** soda-saline soil; cracked soil; spectral features; physicochemical parameters; regression model

## 1. Introduction

Approximately 932 million ha of soil worldwide suffers from primary salinization, and another ~77 million ha from secondary salinization [1,2]. Soil salinization severely impacts the environment and reduces the grain yield. Remote sensing makes it possible to monitor the changes of soda-saline soils in a fast, accurate, and holistic approach. The spectral features of soils are a comprehensive representation of their physicochemical parameters, surface states, and internal structures. It is therefore important to study the spectral reflectance of saline-soils and identify the most sensitive wavelengths in order to classify saline soils [3], and particularly, to measure the salt content and types in soils [4].

During the last century, investigative research into the spectral features of saline soils has been conducted [5–8]. Howari et al. [9] identified the spectral absorption features of soils containing salts including NaCl, NaHCO$_3$, Na$_2$SO$_4$, and CaSO$_4$·2H$_2$O. Farifteh et al. [10] performed laboratory-based assays to determine salinity in aggregate soils based on their spectral features. Pessoa et al. [11] selected soil samples from 78 locations in Pernambuco State and obtained their spectral data. The soil property showing the highest correlation with spectral reflectance was the exchangeable sodium percentage. Matinfar et al. [12] studied the correlation between the soil surface color and the Landsat spectral reflectance of saline soils in Iran, and concluded that the Landsat reflectance in visible light can be used to estimate soil color.

With regard to ground measurements of saline soil spectra, Bellinaso et al. [13] collected the spectral data of 223 soil samples from Brazil using a FieldSpec sensor, and established a ground spectral database, which was used to classify soils. Viscarra Rossel et al. [14] created a spectra database of 5223 soil samples from more than 20 countries. Shi et al. [15] measured the spectra of 1,581 powdered saline soil samples in the laboratory. Spectroscopic predictions of soil organic matter concentrations used a combination of soil spectral classification and multivariate calibrations using partial lease squares regression (PLSR). This combination significantly improved the prediction of soil organic matter ($R^2$ = 0.899, relative percent deviation (RPD) = 3.158) compared with using PLSR alone ($R^2$ = 0.697, RPD = 1.817). Shepherd et al. [16] obtained the spectra of over 1000 powdered saline soil samples from eastern and southern Africa using a FieldSpec FR spectroradiometer (Analytical Spectral Devices Inc., Boulder, CO; USA). The validated $R^2$ using classification trees were the following: exchangeable Ca, 0.88; effective cation-exchange capacity (ECEC), 0.88; exchangeable Mg, 0.81; organic C concentration, 0.80; clay contents, 0.80; sand contents, 0.76; and soil pH, 0.70.

The aforementioned studies showed that soil color, texture, salt content, pH, and clay contents can be predicted from the spectral features of soils. Most soil spectra measurements, however, were previously conducted for powdered or aggregate soils. Under natural conditions, cracking due to dehydration often appears in clay-rich soils, for example, the clay soils on China's Songnen Plain are very sticky and heavy, featuring prominent shrinkage due to dehydration. To date, many studies have focused on the measurement of crack features of clay soils, and how the physicochemical parameters of soils and environmental conditions affect the degree of cracking [17,18]. In another sense, soil crack is a major concern in engineering. Cracks affect the hydraulic and mechanical properties of soils forming landfill liners, slopes, and dams [19]. The crack network considerably modifies the soil's structure and impacts its hydraulic behavior by creating preferential flow paths for fluids and contaminants. Krisnanto et al. [20] proposed a model to predict the lateral flow rate through a network of cracks in the soils. Wan et al. [21] proved that field crack systems may potentially create convectively-driven "hotspots" of enhanced water and carbon gas transport in dryland ecosystems as a result of crack formation.

Soil cracking is very common in nature, and soil salt content is a key property that substantially influences the cracking process [22,23]; however, the spectral features of cracked soils have rarely been studied. To this end, we constructed regression models for the physicochemical parameters of soda-saline soils in different states based on soil spectral reflectance at feature bands. The Songnen Plain, China was selected as the study areas.

## 2. Materials and Methods

### 2.1. Sampling Locations

The Songnen Plain is one of the three largest soda-saline soils in the world. Da'an city, located on the Songen Plain, features the continental monsoon climate, where the mean annual rainfall is only 400–500 mm, but the mean annual reference evapotranspiration is as high as 1400 mm [24]. The hydric deficit, plus the landscape (alluvial and lacustrine plain), the hydrogeological conditions, and human activities, have exacerbated salinization and made this area typical of soda-saline soil. This area was therefore chosen as the study area for our research.

Areas of high soil salt content on the Songnen Plain usually do not have any vegetative cover. As the soil salt content decreases, the amounts of vegetative cover increases. In light of this, we downloaded the thematic mapper (TM) images (http://www.gscloud.cn/?tdsourcetag=s_pcqq_aiomsg) of Da'an in July 2011. From the images, different densities of vegetative cover can be easily distinguished. We collected 57 soil samples of different salt contents at 0–15 cm depth from the surface of the study area (123°42′33″to 124°6′1″E and 45°23′57″to 45°39′57″N) in July 2013 and April 2014. The study area and sampling locations are shown in Figure 1. Soil samples were oven-dried, ground,

and sifted through a 2 mm sieve, and then divided into two groups: one for spectral measurement of powdered soils and the other for controlled experiment of cracking.

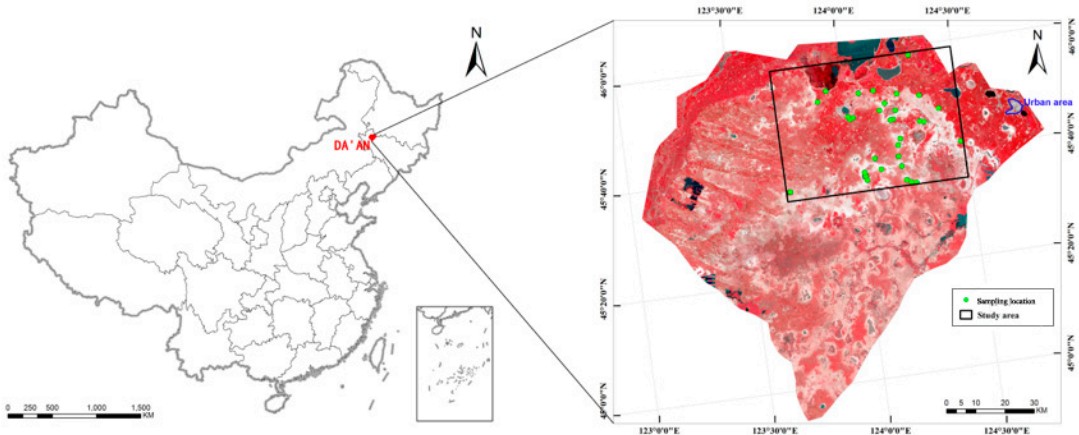

**Figure 1.** Study area and sampling locations.

## 2.2. Soil Parameters Measurement

The soil physicochemical parameters include the contents of eight major ions ($Na^+$, $K^+$, $Ca^{2+}$, $Mg^{2+}$, $SO_4^{2-}$, $HCO_3^-$, $CO_3^{2-}$, and $Cl^-$), pH, electronic conductivity (EC), and particle size distribution. Since the soda-saline soil on the Songnen Plain is almost devoid of $SO_4^{2-}$ [25], this study did not discuss $SO_4^{2-}$ content. For the other parameters, we first made soil extracts with a soil to water ratio of 1:5 and conducted measurements as below:

- $Na^+$ and $K^+$ contents were measured using a flame photometer;
- $Ca^{2+}$ and $Mg^{2+}$ contents were measured using the complexometric ethylene diamine tetraacetic acid (EDTA) titration method;
- $Cl^-$ content was measured using the $AgNO_3$ solution titration method;
- $CO_3^{2-}$ and $HCO_3^-$ contents were measured using the double indicator dilution method [26];
- soil pH and EC were measured using the potentiometric method and conductometric method, respectively.

Then, we measured particle size distribution of the powdered soil samples using a Malvern 115 MS-200 laser instrument.

## 2.3. Soil Cracking Experiment

In order to get cracked soda-saline soils, we treated the 57 soil samples into saturated slurry with the same initial mass water contents, and poured each of them into sample boxes at dimensions of 50 cm × 50 cm × 3 cm. We then smoothed their surfaces and placed the boxes in the laboratory for air drying. The samples were weighed every 12 hours. When the weight of all samples no longer decreased, the air-drying process was considered complete and the formation of cracks stabilized.

To obtain crack features of the soil samples, a digital camera was installed on a platform, facing vertically downward. We used a FUJI FinePix camera, with photo pixel of 3648 × 2432 and bit depth of 24. The camera lens was fixed at 1 m above the ground. A square of 50 cm × 50 cm, with the projection point of the camera lens as the center, was outlined (the same size as the sample boxes). The cracked soil samples were then placed in the square one by one to be photographed. Figure 2 shows the photographs.

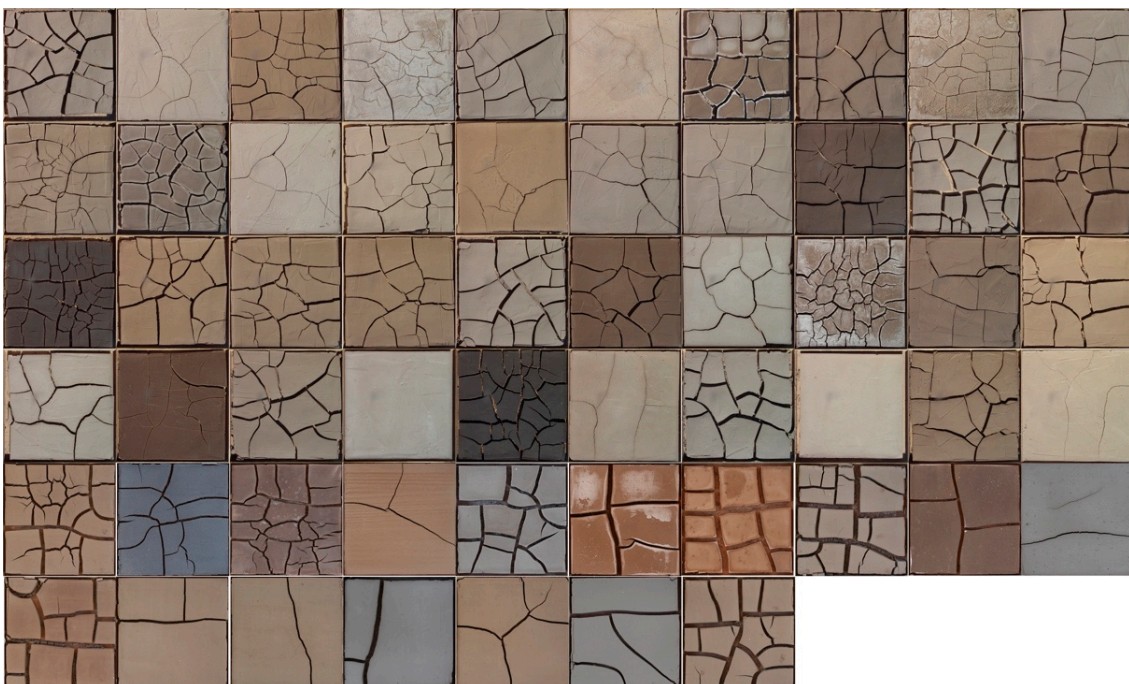

**Figure 2.** Soil samples after air drying (experiment parameters: zenith angle: 0°, distance of capture: 1 m, sample quantity: 57).

### 2.4. Spectra Collection

We collected the spectra of saline soils in powdered, aggregate, and cracked states to analyze and compare their spectral features. A portable, high-resolution spectrometer (ASD FieldSpec 3) was used, with a 350–2500 nm measurement range. There were two detectors: the visible and near-infrared (VNIR) detector collected the spectral reflectance at 350–1000 nm at sampling intervals of 1.4 nm and spectral resolution of 3 nm; the short-wave (length) infrared (SWIR) detector collected the spectral reflectance at 1000–2500 nm at sampling intervals of 2 nm and spectral resolution of 10 nm.

(a)    Spectra collection for powdered soils

Under a clear sky, the sifted powdered soil samples were each placed in an aluminum box of 10 cm in diameter and 1.5 cm in depth—in the optical sense, 1.5 cm is considered infinite thickness. The interior of the sample boxes was painted black to eliminate the impact on spectral reflectance. The surface of the soil samples was smoothened into the same roughness using a ruler. A sensor probe with a field angle of 8° was used for reflectance measurement. It was faced vertically downward, at a distance of 15 cm to the surface of the soil samples (the field of view was a circle of 2.1 cm in diameter, which was smaller than the diameter of the sample box (10 cm)).

(b)    Spectra collection for aggregate soils

A sensor probe with a field angle of 8° was used for reflectance measurement. Under a clear sky, the probe was faced vertically downward at a distance of 15 cm to the surface of the soil samples (the field of view was a circle of 2.1 cm in diameter). We chose a piece of more than 2.1 cm in diameter without cracks out of each soil sample and considered the pieces as aggregate soil samples.

(c)    Spectra collection for cracked soils

A sensor probe with a field angle of 25° was used. Under a clear sky, the probe was fixed onto a platform placed at 1 m above the ground (the field of view was a circle of 45 cm in diameter). The probe was faced vertically downward. A rectangle of 50 cm × 50 cm, with the projection point of the probe onto the ground as the center, was outlined. All cracked soil samples were then placed into this rectangle one by one for spectra collection.

Figure 3 illustrates the scenes of spectra collection for soils in the three states.

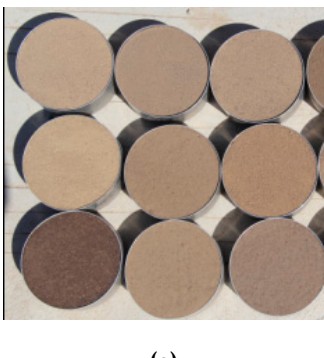 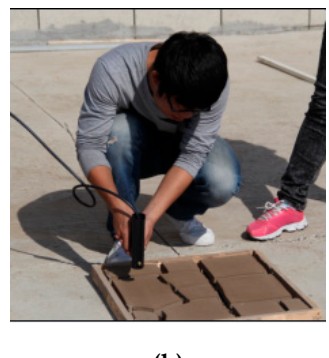 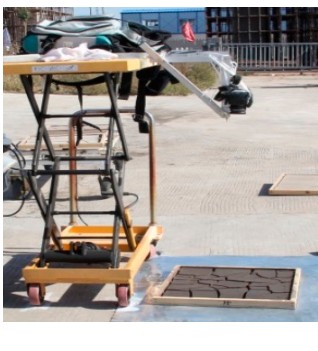

(**a**) (**b**) (**c**)

**Figure 3.** Scenes for spectra measurements of soils in the three states (experiment parameters: (1) field angle: 8°; measurement distance: 15 cm; (2) field angle: 8°; measurement distance: 15 cm; and (3) field angle: 25°; measurement distance: 1 m). (**a**) Powdered soil; (**b**) aggregate soil; (**c**) cracked soil.

*2.5. Spectral Data Processing*

Different wavelengths respond differently to energy, and thus noise existed in the measured spectral reflectance. To remove noise from the spectral data [27], we resampled the data to smoothen the spectral curves. To then truly reflect the curve features such as absorption peaks and valleys, 10 nm was taken as the resampling interval and the Gaussian model was used for spectral resampling. The resampling results were used for calculations in the subsequent steps.

In this process, we coded programs using Matlab, The MathWorks, Massachusetts, MA, USA, (the 2012 version).

*2.6. Crack Ratio (CR) Calculation*

Cracked soil includes cracked parts and aggregate parts. According to the linear mixed spectral model [28], the reflectance of a pixel at a certain wavelength is the linear combination of weighted reflectance of its components, formularized as below:

$$R_b = \sum_{i=1}^{2} f_i c_i + \varepsilon_b; \ \sum_{i=1}^{2} f_i = 1, \tag{1}$$

where $R_b$ is the spectral reflectance at wavelength $b$, $f_i$ is the weighting of component $i$th, $c_i$ is the spectral reflectance of component $i$ at wavelength $b$, and $\varepsilon_b$ is the error. For cracked soils, $i = 2$. $c_1$ stands for the spectral reflectance of aggregate parts, and $f_1$ stands for the ratio of aggregate parts to the entire area. $c_2$ stands for the spectral reflectance of cracked parts, and $f_2$ stands for the CR, which affects the spectral reflectance of cracked soils. CR is calculated by this formula:

$$CR = \frac{A_C}{A_C + A_S}, \tag{2}$$

where $A_C$ is the total area of cracked parts, and $A_s$ is the total area of aggregate parts.

In this study, we used digital image processing to extract the CR of cracked soil samples. Figure 4 takes one sample as an example to illustrate the major steps. First, a polynomial algorithm was used to rectify the geometric distortion of the image, and then the image was cut to keep the cracked sample area only (Figure 4a). The color image was then converted to a gray image (Figure 4b), and a threshold was chosen based on the gray histogram for binarization (Figure 4c), and then inversion was performed (Figure 4d). As a result, the sample's image was segmented into aggregate parts (indicated by black pixels) and cracked parts (indicated by white pixels, including cracks at the four edges of the sample). Erosion was then performed on the binary image to remove noise pixels (Figure 4e). The part of image that maps the field of view (a circle of 45 cm in diameter) of the probe was extracted

to quantify the effect of cracked parts on spectral reflectance (Figure 4f). The ratio of white pixels to the extracted image is the CR of a sample. We implemented this whole process by programming with Matlab (the 2012 version).

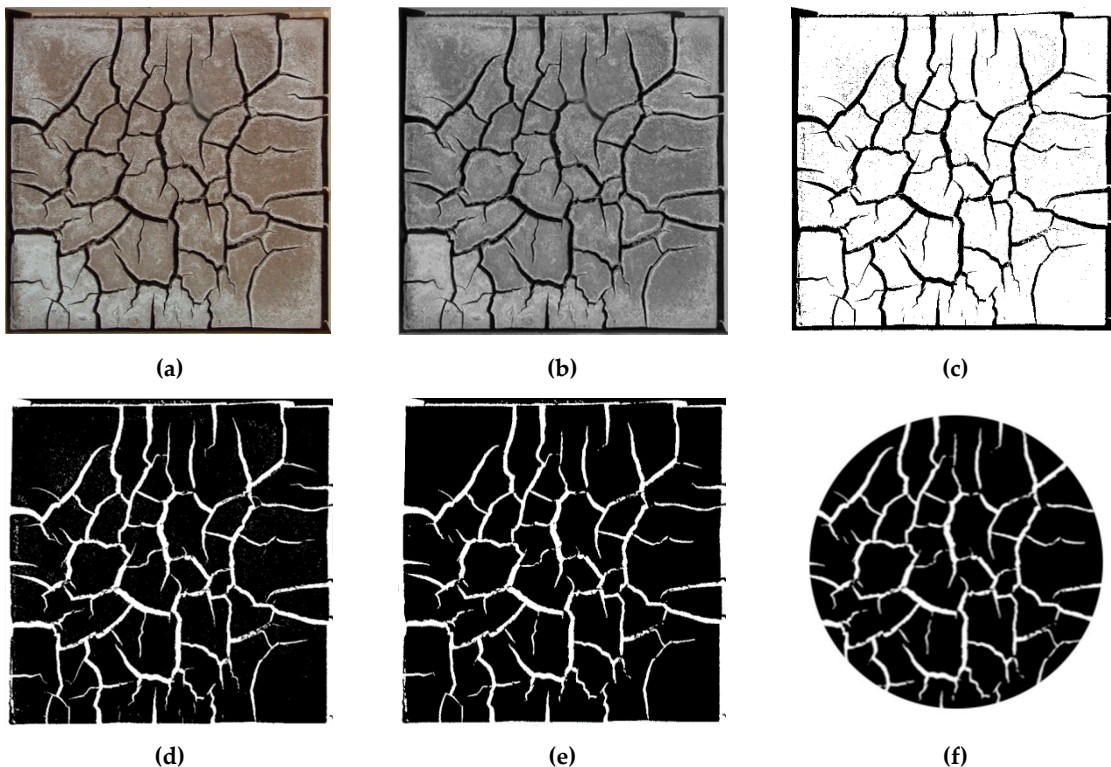

**Figure 4.** Image processing steps. (**a**) Cut original image; (**b**) gray image; (**c**) binarization; (**d**) inversion; (**e**) remove noise; and (**f**) extract image.

### 2.7. Construction of Regression Models for Soil Physicochemical Parameters

### 2.7.1. Construction of Simple Linear Regression Model for Soil Physicochemical parameters

The wavelength with the greatest correlation was selected for establishing a simple linear regression model for the physicochemical parameters. The model is written as $y = ax + b$, where $x$ is the spectral reflectance of a soil sample at the feature wavelength, $y$ stands for any of the physicochemical parameters (including salt content, $Na^+$, pH, and EC), $a$ and $b$ are fitting coefficients that can be determined as described in Section 3.5.1.

### 2.7.2. Construction of the Multiple Linear Regression Model for Soil Physicochemical Parameters

A multiple linear regression model was established between the spectral features of soil samples at four feature wavelengths and their physicochemical parameters. In addition to the feature wavelength selected for the simple linear regression model, three feature wavelengths were selected from the spectral ranges of visible light, near-infrared and shortwave infrared, respectively. The model is written as $y = a_0 + a_1x_1 + a_2x_2 + a_3x_3 + a_4x_4$, where $x_1$–$x_4$ are the spectral reflectance of soil samples at each feature wavelength, $y$ stands for any of the physicochemical parameters (including salt content, $Na^+$, pH, and EC), and $a_0$–$a_4$ are fitting coefficients that can be determined as described in Section 3.5.2.

### 2.7.3. Method for Validating the Models

We calculated the fitting coefficients $a$ and $b$ based on the spectral data of some samples. Then we compared the physicochemical parameters (salt content, $Na^+$, pH, and EC) derived from the models

with the actual parameter values of some other samples. The effectiveness of the models can thus be assessed through the MAE (mean absolute error) according to the formula below:

$$\text{MAE} = \frac{1}{m} \sum_{i=1}^{m} |(y_i - \hat{y}_i)|, \tag{3}$$

where $m$ is the quantity of samples, $y_i$ stands for a physicochemical parameter of sample $i$, $\hat{y}_i$ is the mean of $y_1 - y_i$.

## 3. Results

### 3.1. Measurement Results of the Physicochemical Parameters

Table 1 presents measurement results of the physicochemical parameters for the 57 soil samples, including chemical parameters ($Na^+$, $Cl^-$, $HCO_3^-$, $CO_3^{2-}$, and pH) and physical parameters (EC, salt content, clay, silt, and sand).

**Table 1.** Measurement results of the physicochemical parameters.

| Physicochemical Parameters | Min | Max | Mean | Standard | Coefficient of Variation (CV) (%) | Skewness | Kurtosis |
|---|---|---|---|---|---|---|---|
| $Na^+$ (mg/g) | 0.1 | 14.1 | 3.4 | 3.3 | 97.6 | 1.5 | 2.1 |
| $K^+$ (mg/g) | 0.0 | 0.1 | 0.0 | 0.0 | 64.7 | 2.2 | 5.5 |
| $Ca^{2+}$ & $Mg^{2+}$ (mg/g) | 0.0 | 1.6 | 0.5 | 0.3 | 59.0 | 1.2 | 1.7 |
| Cl (mg/g) | 0.1 | 5.3 | 1.3 | 1.5 | 109.1 | 1.3 | 0.8 |
| $CO_3^2$ (mg/g) | 0.0 | 5.5 | 1.8 | 1.6 | 88.0 | 1.0 | 0.1 |
| $HCO_3$ (mg/g) | 0.2 | 5.0 | 1.6 | 1.0 | 61.7 | 1.2 | 1.4 |
| EC (dS/m) | 0.1 | 3.4 | 1.0 | 0.8 | 84.9 | 1.0 | 0.5 |
| pH (-) | 8.0 | 10.8 | 9.9 | 0.7 | 7.2 | −1.2 | 0.5 |
| Salt content (mg/g) | 1.1 | 29.7 | 8.6 | 6.4 | 74.6 | 1.2 | 1.4 |
| Clay (%) | 25.4 | 32.0 | 28.0 | 1.5 | 5.5 | 0.4 | −0.3 |
| Silt (%) | 28.7 | 40.4 | 35.2 | 3.2 | 9.0 | −0.1 | −0.8 |
| Sand (%) | 28.3 | 43.9 | 36.9 | 3.6 | 9.9 | −0.2 | −0.9 |

$N = 57$.

As the table reveals, the salt content and the contents of major ions differed greatly between the samples. This indicates that the sampling locations are good representatives of the various features of saline soils [29].

### 3.2. Crack ratio (CR) of Cracked Soil Samples

Table 2 summarizes the CR of cracked soil samples calculated according to the method described in Section 2.6.

### 3.3. Soil Spectral Analysis and Determination of the Feature Wavelengths

To perform spectral analysis, the soil sample with intermediate salt content (15.36 mg/g) was selected. Figure 5 shows, as expected, the similar tendencies in spectral reflectance for the three states; where the powdered state showed the highest reflectance and the cracked state showed the lowest reflectance. The figure also illustrates that the reflectance exhibited the absorption feature at 1360 nm and 1900 nm in the near-infrared range, which was due to the absorption of atmospheric water [30], and the reflectance exhibited the absorption feature at 990 nm, 1470 nm, 1990 nm, and 2170 nm, which was due to soil samples containing high concentrations of NaCl and $NaHCO_3$. According to research by Howari [9], NaCl showed the absorption feature at 990 nm and $NaHCO_3$ showed the absorption feature at 1470 nm, 1990 nm, and 2170 nm. Therefore, the results in this study are consistent with Howari's measurements, and 990 nm, 1470 nm, 1990 nm, and 2170 nm were determined as the feature wavelengths.

**Table 2.** Crack ratio (CR) of cracked soil samples.

| Sample No. | 1 | 2 | 3 | 4 | 5 | 6 | 7 | 8 | 9 | 10 |
|---|---|---|---|---|---|---|---|---|---|---|
| CR | 0.136 | 0.013 | 0.082 | 0.041 | 0.067 | 0.011 | 0.141 | 0.095 | 0.039 | 0.04 |
| Sample No. | 11 | 12 | 13 | 14 | 15 | 16 | 17 | 18 | 19 | 20 |
| CR | 0.082 | 0.143 | 0.014 | 0.065 | 0.023 | 0.035 | 0.028 | 0.077 | 0.181 | 0.122 |
| Sample No. | 21 | 22 | 23 | 24 | 25 | 26 | 27 | 28 | 29 | 30 |
| CR | 0.138 | 0.109 | 0.098 | 0.077 | 0.123 | 0.086 | 0.044 | 0.151 | 0.041 | 0.102 |
| Sample No. | 31 | 32 | 33 | 34 | 35 | 36 | 37 | 38 | 39 | 40 |
| CR | 0.071 | 0.029 | 0.11 | 0 | 0.144 | 0.009 | 0.131 | 0 | 0.087 | 0.012 |
| Sample No. | 41 | 42 | 43 | 44 | 45 | 46 | 47 | 48 | 49 | 50 |
| CR | 0.196 | 0.091 | 0.16 | 0.017 | 0.208 | 0.11 | 0.246 | 0.255 | 0.07 | 0.014 |
| Sample No. | 51 | 52 | 53 | 54 | 55 | 56 | 57 | | | |
| CR | 0.201 | 0.037 | 0.02 | 0.056 | 0.049 | 0.087 | 0.23 | | | |

*N* = 57.

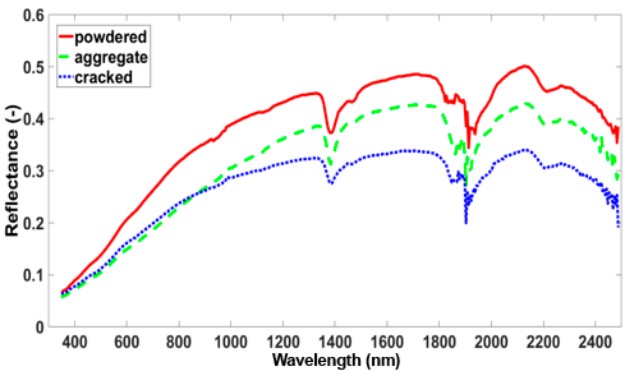

**Figure 5.** Spectral reflectance of a soil sample in three states (salt content: 15.36 mg/g).

Subsequently, we calculated the Pearson correlation coefficients between the spectral reflectance and four of the physicochemical parameters (salt content, Na+, pH, and EC) of the sample in powdered state. As Figure 6 shows, the strongest correlation appeared at 1990 nm, and therefore 1990 nm was determined as the primary feature wavelength.

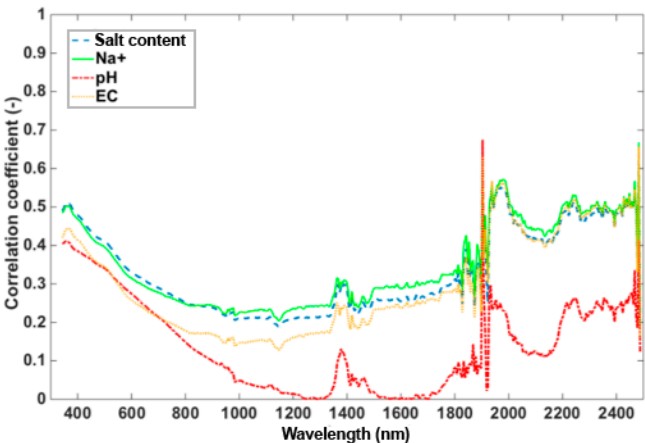

**Figure 6.** Correlation coefficients between the spectral reflectance and four of the physicochemical parameters of a sample in powdered state. Note: correlation coefficient has no unit.

*3.4. Correlation between Soil Spectral Reflectance and Salt Content*

　　We selected six soil samples with salt content of 1.06–29.73 mg/g, at intervals of about 5 mg/g. Figure 7 shows the measured spectral reflectance of these samples in powdered, aggregate, and cracked states. The red curves correspond to the soil sample of the lowest salt content, which showed the highest reflectance; the blue curves correspond to the soil sample of the highest salt content, which showed the lowest reflectance; the same tendency also applied to other curves. It can thus be concluded that the spectral reflectance is inversely related to salt content.

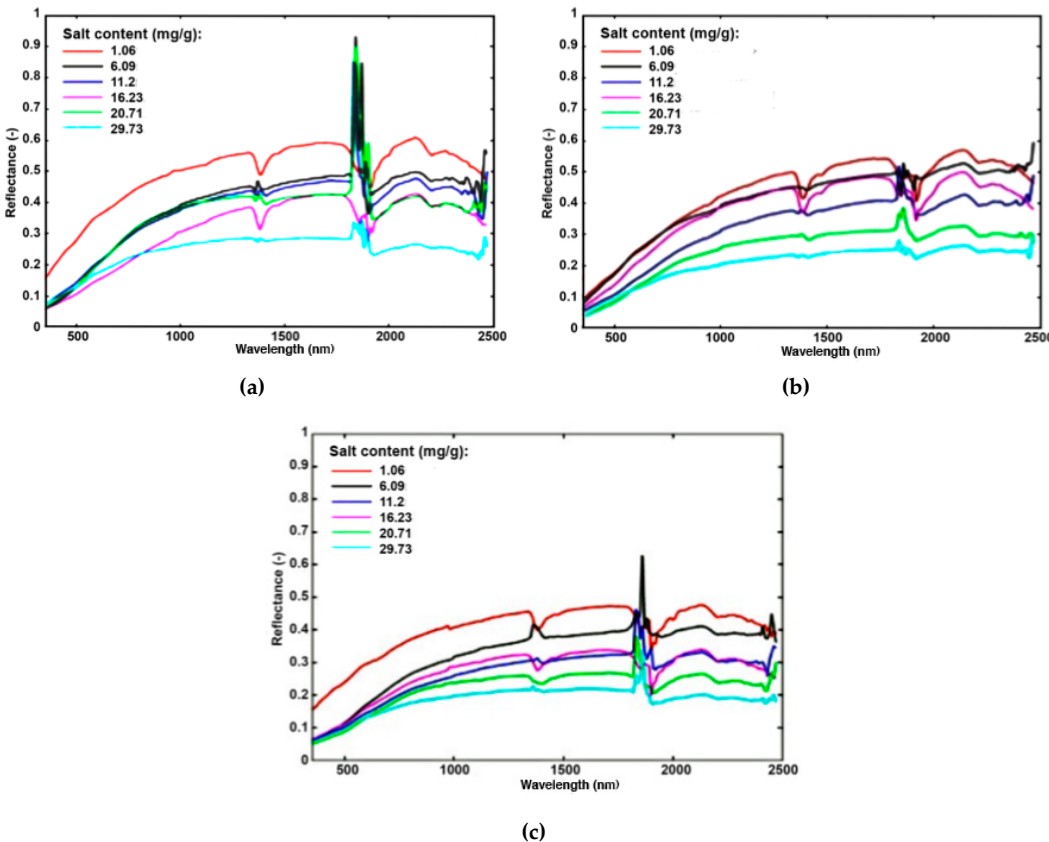

**Figure 7.** Spectral reflectance of soil samples with different salt content. (**a**) Powdered soil; (**b**) aggregate soil; and (**c**) cracked soil.

　　Equation (1) was transformed into $R_p = c_p$ for powdered soils, $R_a = c_a$ for aggregate soils, and $R_c = f_1 c_c + f_2 c_2 + \varepsilon_b = (1 - CR)c_1 + CR c_2 + \varepsilon_b$ for cracked soils. In this study, $c_2 \approx 0$, $\varepsilon_b \approx 0$, and thus the formula for cracked soils was further transformed into $R_c \approx (1 - CR)c_c$. The difference in spectral reflectance was formularized as $\Delta R_p = \Delta c_p$ for powdered soils, $\Delta R_a = \Delta c_a$ for aggregate soils, and $\Delta R_c \approx (1 - \Delta CR)\Delta c_c = \Delta c_c - \Delta CR \Delta c_c$ for cracked soils. As salt content increases, $\Delta CR$ increases, and $\Delta R_c < \Delta c_c$. It can thus be inferred that cracked soils are less sensitive to salt content than powdered and aggregate soils. This is due to the fact that the existence of cracks mitigates the influence of salt content on spectral reflectance.

*3.5. Determination and Validation of the Regression Models*

3.5.1. Determination and Validation of the Simple Linear Regression Model

　　We input the spectral reflectance at the 1990 nm wavelength and the physicochemical parameters of the soil samples into the model described in Section 2.7.1. Then through least squares fitting, the fitting coefficients for four of the physicochemical parameters were determined. The results in Table 3 show that the $R^2$ for cracked soils took on the highest values, indicating that the regression model works more effectively for cracked soils than for aggregate and powdered soils.

**Table 3.** Simple linear regression results for soil physicochemical parameters.

| Physicochemical Parameters | Soil State | $a$ | $b$ | $R^2$ (Determination Coefficient) | Significance Level $\alpha = 0.05$ |
|---|---|---|---|---|---|
| Salt content (mg/g) | Powdered | −44.89 | 29.47 | 0.39 | $2.19 \times 10^{-7}$ |
| | Aggregate | −73.73 | 41.09 | 0.71 | $9.4 \times 10^{-17}$ |
| | Cracked | −77.79 | 36.19 | 0.84 | $2.4 \times 10^{-23}$ |
| Na$^+$ (mg/g) | Powdered | −22.23 | 13.70 | 0.37 | $5.9 \times 10^{-7}$ |
| | Aggregate | −37.66 | 19.96 | 0.73 | $4.4 \times 10^{-17}$ |
| | Cracked | −38.52 | 17.03 | 0.81 | $1.6 \times 10^{-20}$ |
| pH (-) | Powdered | −2.10 | 10.81 | 0.07 | 0.053 |
| | Aggregate | −4.20 | 11.69 | 0.18 | $8.8 \times 10^{-4}$ |
| | Cracked | −5.54 | 11.74 | 0.31 | $6.8 \times 10^{-6}$ |
| EC (dS/m) | Powdered | −5.38 | 3.49 | 0.33 | $3.7 \times 10^{-6}$ |
| | Aggregate | −9.34 | 5.10 | 0.67 | $4.4 \times 10^{-15}$ |
| | Cracked | −9.75 | 4.45 | 0.78 | $3.4 \times 10^{-19}$ |

$N = 57$. The model is written as $y = ax + b$, where $x$ is the spectral reflectance at the 990 nm wavelength.

In order to validate the model, we randomly selected 40 samples to determine the fitting coefficients using the aforementioned method, and calculated the MAE of the other 17 samples. The same process was repeated 50 times. Figure 8 presents the MAE for soils in the three states for all the times. As illustrated, the MAE showed the lowest values for cracked soils, indicating that the regression model achieved the highest accuracy for cracked soils.

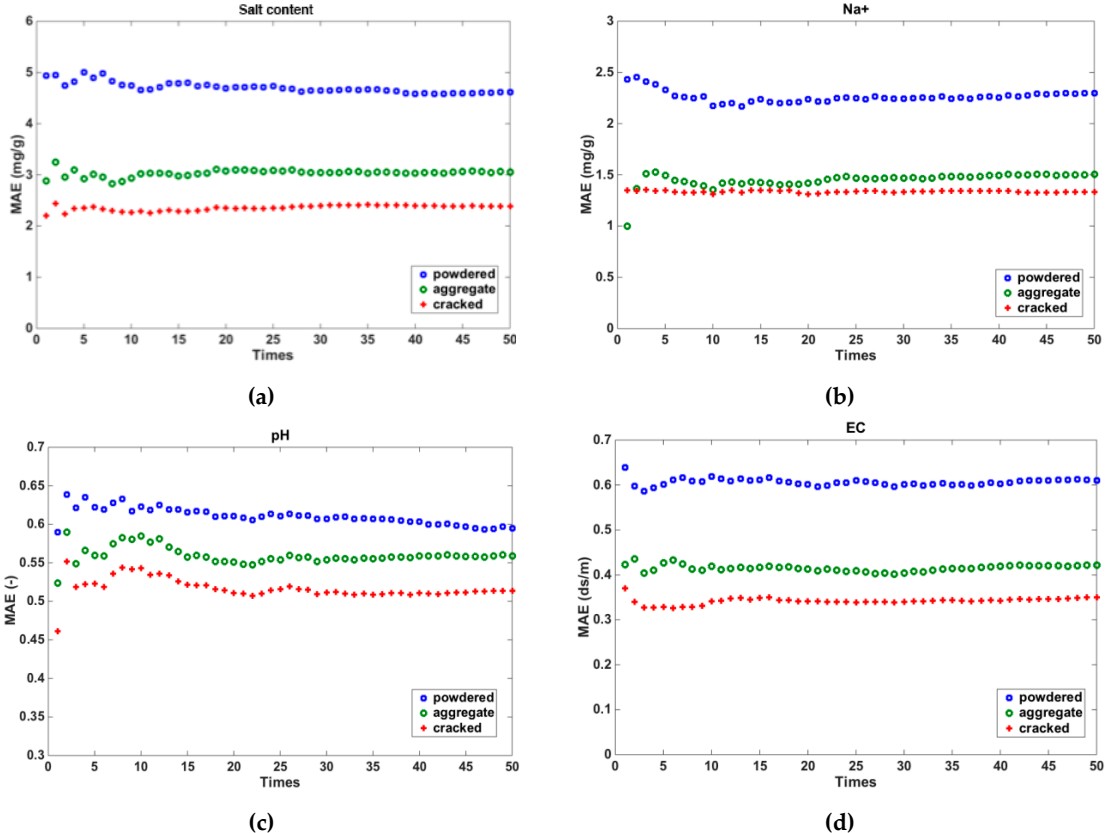

**Figure 8.** Mean absolute error (MAE) derived from the simple linear regression model for soils in different states. (**a**). Mean absolute error for salt content; (**b**). Mean absolute error for Na$^+$; (**c**). Mean absolute error for pH; (**d**). Mean absolute error for EC. Note: since pH does not have a unit, then the MAE of pH does not have a unit either.

In order to further validate the influence of cracks on the accuracy of the model, we introduced the CR parameter into the reflectance of aggregate and powdered soils. The formula for mixed spectral reflectance is as follows:

$$R_b = f_1 c_1 + f_2 c_2 + \varepsilon_b = (1 - CR)c_1 + CRc_2 + \varepsilon_b, \tag{4}$$

where $c_1$ is the reflectance coefficient of aggregate/powdered soils and $c_2$ is the reflectance coefficient of cracks.

With the resulting $R_b$ for powdered and aggregate soils, we ran the validation process again. Figure 9 shows that MAE decreased for modified powdered and aggregate samples, and Table 4 shows that the decrease was 2–11%. This is indication that the accuracy of the regression model improved with the CR parameter considered.

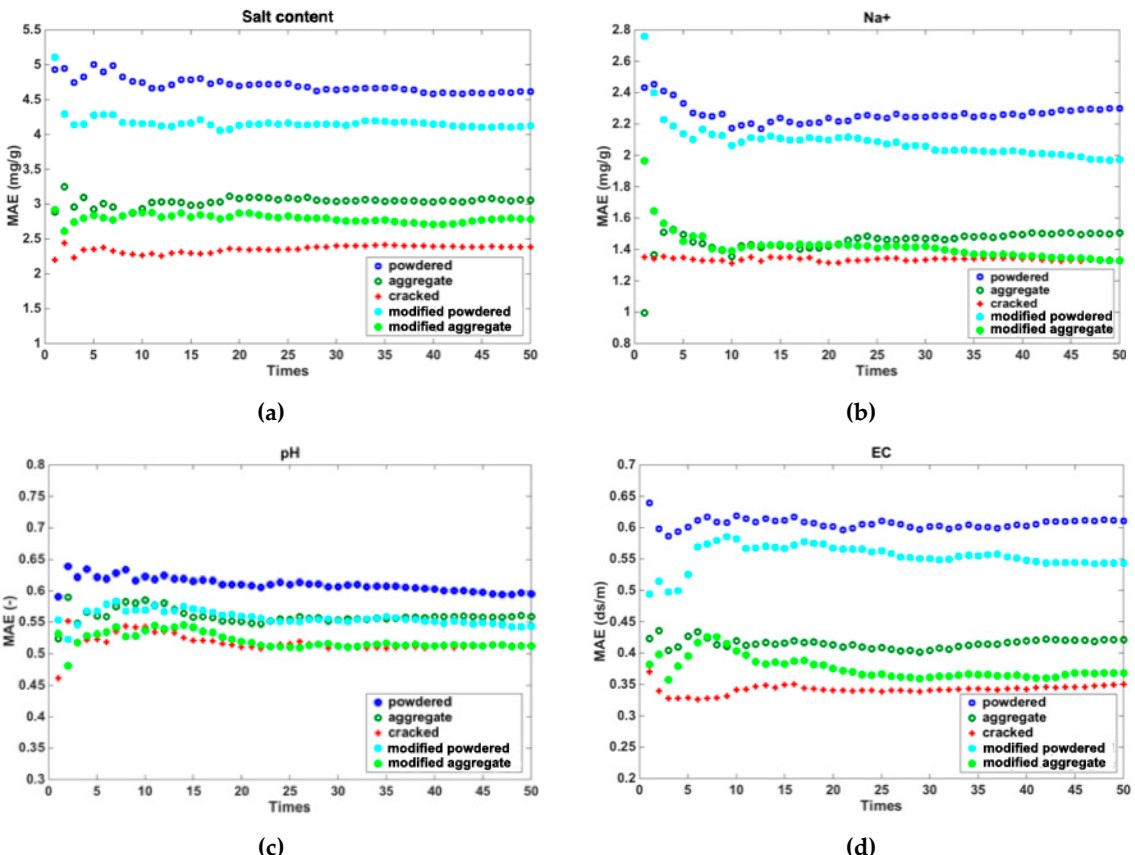

**Figure 9.** Mean absolute error (MAE) derived from the simple linear regression model before and after the crack ratio (CR) parameter was added. (**a**). Mean absolute error for salt content; (**b**). Mean absolute error for Na$^+$; (**c**). Mean absolute error for pH; (**d**). Mean absolute error for EC. Note: since the pH does not have a unit, then the MAE of pH does not have a unit either.

**Table 4.** Decrease of mean absolute error (MAE) derived from the simple linear regression model before and after the crack ratio (CR) parameter was added.

| Physicochemical Parameters | $MAE_{powdered}$ | $MAE'_{powdered}$ (Modified) | $MAE_{aggregate}$ | $MAE'_{aggregate}$ (Modified) | $MAE_{cracked}$ | $\triangle MAE_p$ | $\triangle MAE_a$ |
|---|---|---|---|---|---|---|---|
| Salt content (mg/g) | 4.71 | 4.17 | 3.04 | 2.79 | 2.36 | 11.46 | 8.22 |
| Na+ (mg/g) | 2.26 | 2.08 | 1.45 | 1.42 | 1.34 | 7.52 | 2.07 |
| pH (-) | 0.61 | 0.56 | 0.56 | 0.52 | 0.52 | 8.20 | 7.14 |
| EC (dS/m) | 0.61 | 0.55 | 0.42 | 0.38 | 0.34 | 9.84 | 9.52 |

Number of samples for testing: 40; number of samples for validation: 17; and round of validation: 50. $\triangle MAE_p = \dfrac{MAE_{powdered} - MAE\prime_{powdered}}{MAE_{powdered}} \times 100\%$; $\triangle MAE_a = \dfrac{MAE_{aggregate} - MAE\prime_{aggregate}}{MAE_{aggregate}} \times 100\%$.

### 3.5.2. Determination and Validation of the Multiple Linear Regression Model

We input the spectral reflectance at the four feature wavelengths and the physicochemical parameters of the soil samples into the model described in Section 2.7.2. Then through least squares fitting, the fitting coefficients for four of the physicochemical parameters and the determination coefficient $R^2$ were determined. The results in Table 5 show that $R^2$ improved compared to that determined by the simple linear regression mode. For example, the $R^2$ of $Na^+$ for cracked soils improved from 0.81 to 0.83, the $R^2$ of EC improved from 0.78 to 0.83, the $R^2$ of pH improved from 0.31 to 0.37, and the $R^2$ of the salt content improved from 0.84 to 0.86.

**Table 5.** Multiple linear regression results for soil physicochemical parameters.

| Physicochemical Parameters | Soil State | $a_0$ | $a_1$ | $a_2$ | $a_3$ | $a_4$ | $R^2$ (Determination Coefficient) | Significance Level $\alpha = 0.05$ |
|---|---|---|---|---|---|---|---|---|
| Salt content (mg/g) | Powdered | 2.71 | −35.38 | 214.74 | −105.13 | −106.92 | 0.46 | $2.4 \times 10^{-6}$ |
| | Aggregate | 39.15 | −20.36 | 64.64 | −78.47 | −36.65 | 0.73 | $5.7 \times 10^{-13}$ |
| | Cracked | 33.89 | −19.06 | 59.62 | −180.59 | 61.64 | 0.86 | $6.7 \times 10^{-19}$ |
| $Na^+$ (mg/g) | Powdered | 12.85 | −11.31 | 94.99 | −56.53 | −43.09 | 0.43 | $9.0 \times 10^{-6}$ |
| | Aggregate | 19.87 | 3.4 | 0 | −46.23 | 5.4 | 0.75 | $5.7 \times 10^{-13}$ |
| | Cracked | 15.78 | 1.62 | 11.53 | −126.32 | 73.98 | 0.83 | $1.8 \times 10^{-17}$ |
| pH (-) | Powdered | 10.62 | 0.67 | 12.47 | −12.55 | −1.46 | 0.11 | 0.20 |
| | Aggregate | 11.48 | −4.13 | 3.26 | −3.07 | −0.38 | 0.22 | 0.017 |
| | Cracked | 11.43 | −0.88 | −1.68 | −24.84 | 21.62 | 0.37 | 0.00011 |
| EC (dS/m) | Powdered | 3.16 | −3.83 | 30.46 | −19.25 | −10.82 | 0.41 | $2.0 \times 10^{-5}$ |
| | Aggregate | 4.8 | −0.25 | 4.13 | −17.77 | 4.76 | 0.68 | $2.0 \times 10^{-11}$ |
| | Cracked | 3.89 | −1.88 | 10.74 | −42.14 | 23.34 | 0.83 | $1.2 \times 10^{-17}$ |

$N = 57$. The model is written as $y = a_0 + a_1x_1 + a_2x_2 + a_3x_3 + a_4x_4$, where $x_1$–$x_4$ stand for the reflectance at the 990 nm, 1470 nm, 1990 nm, and 2170 nm wavelengths.

In order to validate the model, we followed the same procedure as we did for validating the simple linear regression model. As presented in Figure 10, the MAE showed the lowest values for cracked soils, indicating that the multiple linear regression model achieved the highest accuracy also for cracked soils.

Like what we did with the simple linear regression model, we also introduced the CR parameter into the reflectance of powdered and aggregate soils. Figure 11 shows that the MAE decreased for modified powdered and aggregate samples.

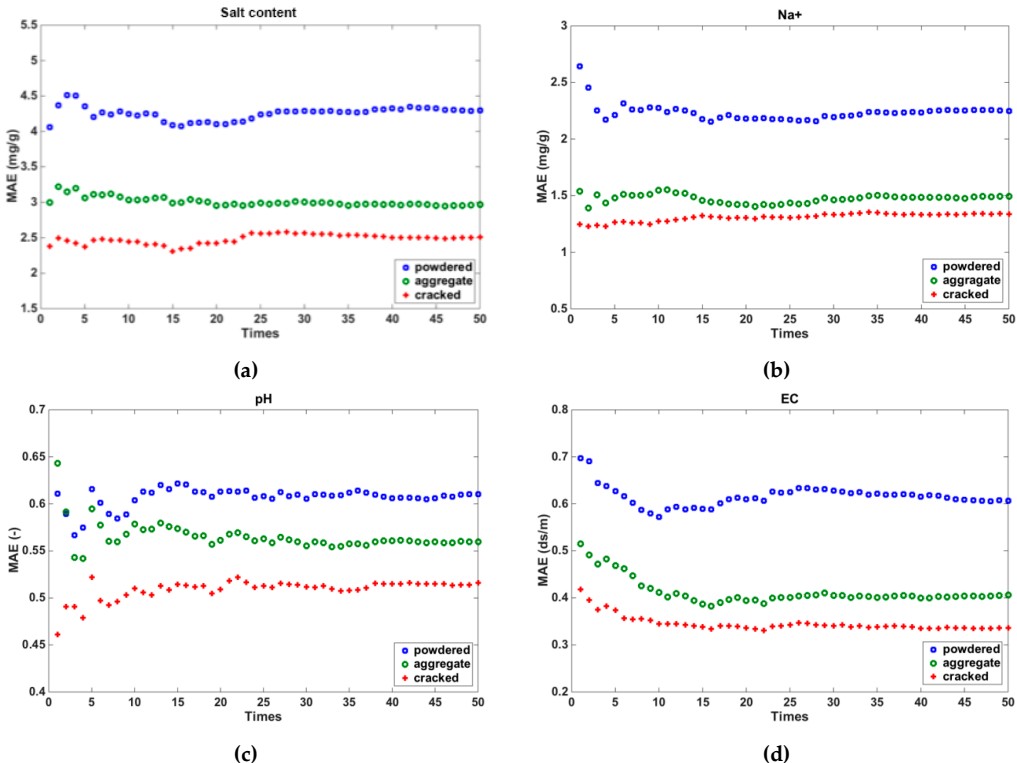

**Figure 10.** Mean absolute error (MAE) derived from the multiple linear regression model for soils in different states. (**a**). Mean absolute error for salt content; (**b**). Mean absolute error for Na$^+$; (**c**). Mean absolute error for pH; (**d**). Mean absolute error for EC. Note: since pH does not have a unit, then the MAE of pH does not have a unit either.

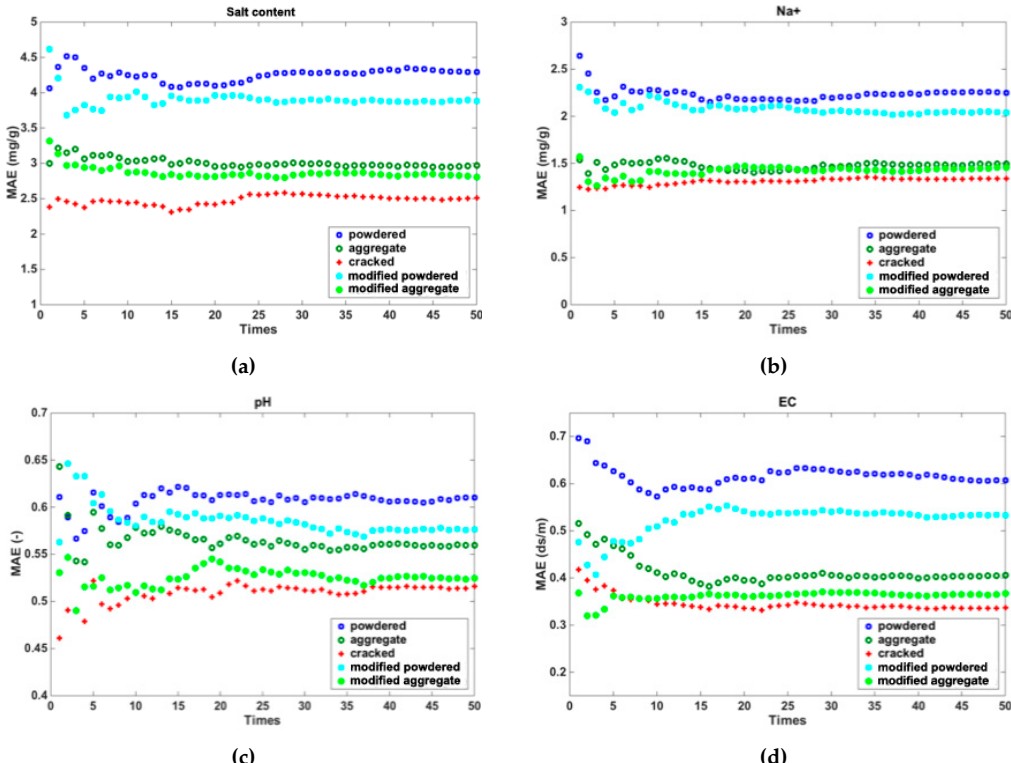

**Figure 11.** Mean absolute error (MAE) derived from the multiple linear regression model before and after the CR parameter was added. (**a**). Mean absolute error for salt content; (**b**). Mean absolute error for Na$^+$; (**c**). Mean absolute error for pH; (**d**). Mean absolute error for EC. Note: since pH does not have a unit, then the MAE of pH does not have a unit either.

## 4. Discussion

In general, spectral reflectance is influenced by the soil moisture, salt content, surface roughness, organic carbon, etc. [31]. As shown by previous research, spectral features of soils are in inverse exponential correlation with soil moisture [32–34]. This means that when the moisture content increases, the spectral reflectance decreases. Researches also showed that the soil reflectance decreases with soil salt content [35,36]. In the current study, we took dried soil samples as the study object, which can be considered to have the same moisture content, and the main factors that determine the spectral reflectance are salt content and surface roughness. For a certain soil sample, the salt content was the same in different states, leaving surface roughness as the single determining factor for spectral features. Powdered soils, having the smoothest surface, showed the highest reflectance since the powder particles are all smaller than 0.9 mm in diameter; aggregate soils are less smooth since particles stick to each other more tightly; and cracked soils are the least smooth because of cracks.

Soil cracking is a complicated mechanical process, which is influenced by soil mineralogy, type, texture, and organic carbon contents, among other factors. Research has shown that cracking of clayey soil is largely determined by its mineral types and clay contents [37,38]. Soda-saline soils on the Songnen Plain contain almost the same minerals and clay minerals. This is because soil-forming conditions, including the geographical environment, the ecological environment, the climatic condition, and the hydrologic process, are all stable on the Songnen Plain. Table 1 reveals that for clay, whose particle size is less than 0.005 mm, the standard deviation and variation coefficient are only 5.5% and 1.5%, respectively. This indicates that soil samples in this study accommodate almost the same clay content. Research by Zhang et al. [39] showed that the clay content of soda-saline soils on the Songnen Plain ranges from 20% to 29%, but the activity index ranges only from 0.33 to 0.48, which puts the soils into the no-activity category. This is because clay contents are mostly illite/smectite mixed layer (more than 50%), along with some ledikite and kaolinite, all of which are nearly not active. Meanwhile, there is almost no highly active smectites in the soils. Thus, the clay content will not lead to desiccation [40,41]. However, desiccation is still common on the Songnen Plain under natural conditions. This is due to the dehydration process, in which the soil particles interact with exchangeable cations (particularly Na+, which has a large hydrolytic radius and was the main cationic component of the soda-saline soil) within the soils, forming a relatively thick-bonded water film between soil particles. This bonded water film was thicker for soils of higher salt content, and it reduced the cementation and increased the distance between soil particles, resulting in decreased soil cohesion and tensile strength [42,43]. In addition, the lubricating effect of the bonded water film between soil particles reduced the internal friction angle between soil particles as well as the shear strength of the soil [44,45]; therefore, soils of higher salt content exhibited more prominent cracks.

The spectral reflectance shows similar tendency for a soil sample in three different states, with the feature wavelength appearing at the same position (Figure 5). It is also apparent in the figure that the powdered state shows the highest reflectance, followed by the aggregate state, and finally the cracked state. The difference is caused by surface roughness since the soil contains the same salt content in whatever state. As surface roughness increases, the diffuse reflection becomes stronger, and the probe receives less energy. This is aligned with research by Leblon et al. [46] and Zhai et al. [47], who analyzed the effect on the visible and near-infrared reflectance of shadows on different surfaces. The results showed that the reflectance of shadows is significantly affected by the surface type and the shadow size. In this study, when the sample is in the cracked state, the existence of cracks increased the surface roughness, and moreover shadows appeared because sunlight was not vertical during spectra measurement. As a result, the reflection energy is close to zero in cracked areas and the cracked state shows the lowest reflectance.

For soil samples in a certain state, reflectance is mainly influenced by salt content (Figure 7). Specifically, reflectance decreases as salt content increases, which is particularly evident at wavelengths greater than 1400 nm. This agrees with the research by Wang et al. [48], Farifteh et al. [49–51], and Weng et al. [35]. Moreover, the spectral reflectance became flatter with further increase in soil salt

content [52]. The reflectance of soils in different states shows an increasing trend at 350–1400 nm. Whereas the increase slows down at 1400–2200 nm, and a decreasing trend appears at 2200–2500 nm. The cause lies in $NaHCO_3$ in the soil, whose reflectance showed an evident decreasing trend at wavelengths greater than 1400 nm. Moreover, as the salt content increases, the trends become more evident. CR exerts more influence on spectral reflectance than the particle size for cracked soils (Figure 7). For powdered and aggregate soils, the prediction accuracy of the regression models improved when the CR parameter is added (Figures 9 and 11).

The spectral features obtained by satellites are mostly for cracked saline soils [53–55], since cracking is prevalent on saline soils due to water evaporation in natural conditions. In case when only powdered or aggregate soils are measured, the accuracy of the spectral inversion for soil salt content can be improved by adding CR into the spectral mixing models.

Sampling locations in this study are limited to only part of the Songnen Plain. Factors such as mineral type, soil fabric, and organic carbon were not studied. In future studies, the influence of these factors on spectral reflectance can be analyzed so that the study results can be applied to more areas.

## 5. Conclusions

In this study, we selected saline soils from Da'an, on the Songnen Plain. Fifty-seven soil samples of different salt content were collected and their spectral reflectance was measured in powdered, aggregate, and cracked states. Then a simple linear regression model and a multiple linear regression model were constructed between the spectral reflectance and the physicochemical parameters (salt content, $Na^+$, pH, and EC) for saline soils in different states. The models were also validated by comparing the prediction results with the measured actual values. The following conclusions are yielded from this study:

(a) The spectral reflectance showed similar tendency for a soil sample in different states, with the feature wavelength appearing at the same position, but the powdered state showed the highest reflectance, followed by the aggregate state and lastly the cracked state.

(b) or soils in the same state, reflectance was inversely proportional to salt content.

(c) The regression models between the spectral reflectance and the soil physicochemical parameters work more effectively for cracked soils than for aggregate and powdered soils.

(d) With the CR parameter added into the spectral mixing models, the MAE decreased by 2–11%, a clear indication that the prediction accuracy improved.

**Author Contributions:** Conceptualization, Z.L. and K.Z.; methodology, X.L.; software, J.R.; validation, X.L. and K.Z.; formal analysis, J.R.; investigation, X.L. and Z.L.; resources, Z.L.; data curation, J.R.; writing—original draft preparation, X.L.; writing—review and editing, X.L.; visualization, X.L. and J.R.; supervision, K.Z.; project administration, X.L.; and funding acquisition, X.L.

**Funding:** This research was funded by the National Natural Science Foundation of China (No.41671350) and the Excellent Young Talents of Jilin Province (No.20170520090JH).

**Conflicts of Interest:** The authors declare no conflict of interest.

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
