# Peer review of "Correlation between Spectral Characteristics and Physicochemical Parameters of Soda-Saline Soils in Different States"

_remotesensing, doi:10.3390/rs11040388_

Round 1

Reviewer 1 Report

remotesensing-432055

I went through the manuscript entitled as "Correlation between spectral characteristics and salt parameters of soda-saline soils in different states”. This manuscript deals to build a correlation between spectral and physiochemical properties of saline soils. The manuscript is well designed and the paper is basically correctly written. Comparing to the previous version great improvement happened. However, there are some issues that they are listed below:

L93: “salinity at 0 ~ 15 cm deep from” change to “salinity at 0 ~ 15 cm depth from”

L178: it would be better to write the formula in the dependent line.

L 168, 203: Please add the version of Matlab software.

L301-302: this sentence repeated. Remove it.

Table 4: The first row of the table made this table confusing. It is better to remove the formulas from this row and write them as caption below the table.

References:

Is the references provided in the right format (Journal style). There are some differences such as below:

Hunt, G.R., Salisbury, J.W., Lenhoff, C.J. Visible and near-infrared spectra of minerals and rocks. IV. Sulphides and sulphates. Modern Geology 3, 1971, pp 1–14

H.R. Matinfar, S.K. Alavipanah, F. Sarmadian. Soil spectral properties of arid region, Kashan area, IRAN. BIABAN Journal. 2006, 11(1), pp 9-17. doi: 10.22059/jdesert.2006.31862.

So it advise to check all again.

Author Response

Dear reviewer,

We would like to thank you for your constructive comments. We have carefully revised our manuscript according to your comments. The revised manuscript was uploaded with all the changes highlighted. Please note the line numbers refer to those in the final version. Below you can find our response to the comments point by point:

I went through the manuscript entitled as "Correlation between spectral characteristics and salt parameters of soda-saline soils in different states”. This manuscript deals to build a correlation between spectral and physiochemical properties of saline soils. The manuscript is well designed and the paper is basically correctly written. Comparing to the previous version great improvement happened. However, there are some issues that they are listed below:

L93: “salinity at 0 ~ 15 cm deep from” change to “salinity at 0 ~ 15 cm depth from”

Thanks. We have revised it according to your suggestion. Please see line 95.

L178: it would be better to write the formula in the dependent line.

Thanks. We have revised it according to your suggestion. Please see line 181-182.

L 168, 203: Please add the version of Matlab software.

OK. We have revised it according to your suggestion. Please see line 170.

L301-302: this sentence repeated. Remove it.

Thanks. We have revised it according to your suggestion. Please see line 311-312.

Table 4: The first row of the table made this table confusing. It is better to remove the formulas from this row and write them as caption below the table.

Thanks. We have revised it according to your suggestion. Please see line 327.

References:

Is the references provided in the right format (Journal style). There are some differences such as below:

Hunt, G.R., Salisbury, J.W., Lenhoff, C.J. Visible and near-infrared spectra of minerals and rocks. IV. Sulphides and sulphates. Modern Geology 3, 1971, pp 1–14

H.R. Matinfar, S.K. Alavipanah, F. Sarmadian. Soil spectral properties of arid region, Kashan area, IRAN. BIABAN Journal. 2006, 11(1), pp 9-17. doi: 10.22059/jdesert.2006.31862.

So it advise to check all again.

Thanks for your suggestion. We have revised them.

Thanks again for your constructive review!

Xiaojie Li, Kai Zhao, Jianhua Ren, Zhengwei Liang

January 26, 2019

Reviewer 2 Report

Revision of the manuscript remotesensing-432055-peer-review-v1 “Correlation between spectral characteristics and salt 2 parameters of soda-saline soils in different states”.

In general: Please be careful, and use properly the standard/appropriate terminology for soil analysis methods, remote sensing, and vegetation and other descriptions and analysis. Check it and also the narrative through the manuscript. Authors should check and explain the methods in relation with the existing literature.

A native English edition (scientific expertise) of the text is strongly advisable.

Comments to the authors

L30. “these “ It's not understood.

L31. Remove “ecological”. There is no sense to qualify environment here.

L60 Remove “level”. What do mean with “soil type”, soil type in the sense of taxonomic classification is not probable.

L74 Use salt content instead of salinity, here.

L78 “explored in depth the spectral characteristics in these states” is repetitive. Use analysed instead of measured

L79. “salt parameters” Do you refer to salt content /composition?

L82 I do not understand “concentrated”

L84 It is understood that a city (urban area?) is selected as study area.

L85, “evaporation”: as climatic variable please specify if it is mean annual reference evapotranspiration (ET0) or potential evapotranspiration (ETP).

L86: Remove “between rainfall and evapotranspiration”,  it is redundant

L87. I do not understand “sharp bulges” as a landscape / geomorphological element. Please, explain how the landscape conditions soil salinity. Remove “special”, use a more precise description if necessary.

L91. “of different density”, I do not understand. Do you mean Different vegetation cover?. Please, clarify. Downloaded from where? how many images?, and which dates were analysed? What is the most relevant information to understand the methodology?

L92-93. It is preferable use “vegetation” instead of “plants”. Capital letter “We”.  “… samples in areas of different vegetation cover, corresponding to soils with different salt content…” the sentence  would be more fair like this. How many images, dates, were used for photointerpretation /Visual analysis (this is the method I understand was used and should be clarified: which bands used for this task?) I do not understand how you can select punctual areas for soil sampling from a 30-m pixel size satellite images taking into account that 1) halophytes cover, especially annual pioneering plants can be really small and are scattered, 2) the pixel offer you an average reflectance, a mixed response, especially depending the bands used for interpretation; 3) salt content, especially efflorescence or salt crusts, saturate the sensor and mask the reflectance of non-white components in the soil surface (avoiding punctual plant identification); this is described in litterature. I guess visual analysis and interpretation might have helped to select broad areas of assumed mostly bare soil. But point selection must have been a field task based on other detailed /site specific criteria. Please explain properly, it does not need to be extended in the explanation but clarity.

L96 To my knowledge sieves have not pores but mesh light. Again, revise and use correct technical /scientific terms through the manuscript.

L102. EC is never used with  “s” . Particle size distribution. These and other terms are standards that are found in the literature and any manual for a long time.

L103 Please, add in study area section the general lithology /mineral / or general composition or whatever was necessary to understand here the lack of SO4 in the saline soils studied.

L104 Ion content. The standard is 1:5 soil:water extract .

L110. Add reference or clarify the method

L112. EC is never used with s.

L114. Remove “And”. Particle size distribution.

L115. Malvern?

L128. The photographs are shown

L132. In my opinion you are studying three different soil treatments. I understand that “states” usage is not appropriate here.

L167. Please, check and explain if the noise removal method you used is similar or different from those described in the literature.

L204. Salt parameters (throughout all the manuscript): do you refer to salt composition?. EC is not a salt parameter, but a soil characteristic or feature. Please correct.

L225-226. Rewrite green sentence please. Moreover: salinity measured as EC is a hysical parameter. Correct in all the manuscript

L228-229. Data in the table has only one decimal. Again, corrected I the previous revision. EC unit is dS/m. Siemen, capital letter.

Since you are using two determination methods for soil salinity description (mg/g and electrical conductivity), you should justify why, and use “salt content” for the first method (mg/g), to differentiate them.

“which indicates that all soil samples exhibited apparent characteristics of alkaline”. Alkaline soils. I do not understand this statement, I suppose that alkalinity refers to pH >8. Concerning salinity, the authors must check their EC values of soil salinity, with soil salinity intervals (non-saline, saline, … to strongly saline) from literature (US standard classes of soil salinity, etc.).  A priori EC=0.1 corresponds to a non-saline soil but it is improbable since you selected bare soils assumed highly saline. Please explain and discuss.

L233 Idem. Size.  

L234 You should add the main textural class (standard USDA textural triangle, for example)

L270 Red curves /red spectra correspond to soils….. “Soil salinity” should be better the caption of the legend instead of repeated..

I am sorry. The revision of the following sections requires formal prior review of all these issues throughout the manuscript.

Author Response

Dear reviewer,

We would like to thank you for your constructive comments. We have carefully revised our manuscript according to your comments. The revised manuscript was uploaded with all the changes highlighted. Please note the line numbers refer to those in the final version. Below you can find our response to the comments point by point.

For the details please see attachment.

Thanks.

Reviewer 3 Report

The article shows the effect the surface state of soda-saline soils have on their spectral characteristics and relationships between surface spectral reflectance and salt parameters. Surface state is significant factor which should be considered when modelling soil properties form spectral data. Therefore this research deals with very important but hardly studied issue in soil remote sensing.

I do have some comments I would like authors to address. They are given below.

line 22 - usually regression model has certain accuracy (not precision)

Also I would like to ask authors to formulate the main aim of their research in introduction.

Line 180 - I think the explanation should be corrected. As far as I understood, the first equation is for linear spectral mixture  and the second one shows that sum of the weights of the componets of linear mixture should equal one. Why is it said that these two formulas for soils and cracks?

Lines 182-184 I would disagree with this statement. It is too strong. The equation only shows that CR affects spectral reflectance as cracks are the component of spectral mix. The importance and degree to which CR contributes to spectral reflectance should be proved. What is done further in the study. 

line 205 - missed letter: linear

line 222 - give the formula for mae

table 2 - provide units of measurments (is it % divided by 100?)

line 274- you mean the reflectance of cracked soils is more sensitive to salinity?what is the foundation for this conclusion?

line 303 - how and where exactly CR parameter was added? provide an example or an equation.

does it actually make sense to add CR to powdered state? Can this happen in the field?

line 323 - there is no correlation coeffitien in table 5. however, there is determination coefficient

table 5 -why were the same wavelenghts included in all the models for all the properties? Were they predetermined before the regression analysis? 

line 437 - regression model (not correlation)

Author Response

(The authors gave the same response as above.)

Round 2

Reviewer 2 Report

Comments to the manuscript remotesensing-432055-peer-review-v2.

L105. Add the reference after SO4 . Liu Qiang , Baoshan Cui, Zhifeng Yang. Dynamics of the soil water and solute in the sodic saline soil in the Songnen Plain, China. Environ Earth Sci, 2009, 59:837-845. DOI:10.1007/s12665-009-0079-4.

L116. Replace “a Malvern 115 MS-200 laser particle size distribution” with “a Malvern 115 MS-

200 laser instrument”

L171. Add the reference for noise removal method used. R K. SAHA, B. B. CHAUDHURI, D. DUTTA MAJUMDER. A NEW SHAPE PRESERVING PARALLEL THINNING ALGORITHM FOR 3D DIGITAL IMAGES. Pattern Recognition, 1997, 30(12), pp 1939-1955.

L299 “Salt parameters” is still uncorrected in the manuscript and asked for correction in the previous review.

L204. Salt parameters (throughout all the manuscript): do you refer to salt composition?. EC is not a salt parameter, but a soil characteristic or feature. Please correct.

Thanks for your suggestion. We have replaced it by “soil physiochemical parameters” and rewritten this sentence. Please see line 235-237.

L240 …of alkaline soils. Here you should use “salt content” instead of “salinity” since “EC” is a international standard salinity measure in soils (even more than mg/g). For this reason you should clarify at the beginning of the methods that you will use the term salinity for salt content in mg/g or revise the use of “salinity” in all the manuscript to avoid this misunderstanding. This issue was asked to be corrected/ clarified in the previous review.

You must use a soil salinity standard reference, other than Szabolcs, 1974. It is not standard and out of use,

Moreover, your table 1 in the cover letter includes salinity as % and you are using g/mg. Clarify / correct comparisons.

FAO reference salinity classification is using EC from saturated paste extracts, which is really the standard classification but you have not these EC values but 1:5 extractcs. They are not comparable.

 Reference 27: author used criteria of the U. S. Salinity Laboratory lab: the salinity of saturated paste extract (ECe) (You always should use the original reference 

Richards , L. A. , ed. 1954 . Diagnosis and Improvement of Saline Alkali Soils . USDA Handbook No. 60 . Washington , DC , USA

L247. 1.5%

L301 Add parameter units in Table 3. Salinity (mg/g), EC (dS/m) and add the unit of Na. Add also a heading for this column

L254. Simplify subheading : soil spectral analysis, for example

L256 soil sample A soil sample with intermediate salt content ( xxx) was selected.

L255-256. First sentence can be removed. Figure 7 is evident

L258… Rewrite, please.  ..as expected, the reflectance of the three samples is similar in shape

L259 Please, improve the narrative to make it technically sound. … the reflectance values of the … were higher than /the highest… and the aggregate state showed intermediate reflectance values.

L260-262 Remove curves. I do not understand apparent. You should add a reference for “the effect of atmospheric moisture”. Indicate the name of the waveband interval 1360 nm and 1900 nm.

L261 to 264. Reword please.

L265 Correct  adsorption.

L266. I do not understand if these are your results of are the typical absortion features mentioned in literature. Please clarify.

L264: “minerals” is not correctly used. Do not cause confusion between minerals and salts, sodium chloride versus halite. Idem for L267 and all the manuscript. Uncorrected from previous review.

L273. 1990 nm

L273 legend: salt content (g/mg)

L279.  … six soil samples with salt content values from xxx to xxx g/mg, at intervals …. Or adopt other appropriate writing

Revise from the previous review: L116 and 118. I understand that the symbol used (approx.) should be a hyphen (interval)

Corrections from previous review are uncomplete.

Author Response

Dear reviewer,

We would like to thank you for your constructive comments. We have carefully revised our manuscript according to your comments. Please note we have uploaded two versions of the manuscript in one document. The first one shows all markups and the second is the clean version. The line numbers below refer to those in the clean version. Please find our response to the comments point by point.

Thanks again for your constructive review! 

Xiaojie Li, Kai Zhao, Jianhua Ren, Zhengwei Liang

February 8, 2019

Reviewer 3 Report

It is good to see the answers to my comments. However, expalnations should also be included in the manuscript. Regarding my comment to line 303 about the addition of CR parameter (line 319 in the revised manuscript). Please include your explanation in the manuscript. I did not ask this question just out of curiosity. This information will help other researchers to use your methodology in their studies. 

The same goes for line 274 (line 284-285 in the revised manusrcipt). Actually, I will advise to drop this sentence or give a good evidence of that with your data. You did not show the relationship between CR and salinity. Your statement is not supported by the results you presented. 

Please use concise terminology. You either analysed different soil treatments or soil states. Just stick to one.

Table 2 - May be CR is uniteless as  it is a ratio (according to the equation 2)? In this case you need to remove 100%. Because it looks strange.

Author Response

Dear reviewer,

We would like to thank you for your constructive comments. We have carefully revised our manuscript according to your comments. Please note we have uploaded two versions of the manuscript in one document. The first one shows all markups and the second is the clean version. The line numbers below refer to those in the clean version. Please find our response to the comments point by point:

Comments and Suggestions for Authors

It is good to see the answers to my comments. However, expalnations should also be included in the manuscript. Regarding my comment to line 303 about the addition of CR parameter (line 319 in the revised manuscript). Please include your explanation in the manuscript. I did not ask this question just out of curiosity. This information will help other researchers to use your methodology in their studies. 

Thanks for your suggestion. We have added them in manuscript. Please see line 296-301.

The same goes for line 274 (line 284-285 in the revised manusrcipt). Actually, I will advise to drop this sentence or give a good evidence of that with your data. You did not show the relationship between CR and salinity. Your statement is not supported by the results you presented. 

Thanks for your suggestion. Sorry that we did not descript it clear enough. We have revised them. Please see line 261.

Please use concise terminology. You either analysed different soil treatments or soil states. Just stick to one.

Thanks for your suggestion. We have used “soil states” through the whole manuscript.

Table 2 - May be CR is uniteless as it is a ratio (according to the equation 2)? In this case you need to remove 100%. Because it looks strange.

Thanks for your suggestion. You are right that CR is unitless as it is a ratio. We have deleted it. Please see line 233.

Thanks again for your constructive review! 

Xiaojie Li, Kai Zhao, Jianhua Ren, Zhengwei Liang

February 6, 2019

This manuscript is a resubmission of an earlier submission. The following is a list of the peer review reports and author responses from that submission.

Round 1

Reviewer 1 Report

The article is devoted to the modelling of relationships between soil spectral characteristics and soil salinity. My comments and suggestions can be found below.

In subsection 2.4 Processing of spectral data, please, specify the software you used for preprocessing.

In subsection 2.5, please, specify the software you used to analyse photo images.

In subsection 2.6 you refer to correlation model (as well as in the heading). However, I believe, it is regression modelling you are doing. There is correlation coefficient, correlation analysis, but no correlation modelling. If otherwise, please, provide a reference. So,  in 2.6.1 you desribe simple regression modelling and in 2.6.2  - multiple regression. 

In Results in subsection 3.1 you describe the results of the laboratory analysis, but in Methods section you did not mention what laboratory methods were used.

line 227 - adsorption (absorption????)

line 237 - it is not clear why you chose powdered state to determine the primary characteristic waveband?

line 248 - when you speak about positive correlation it is not clear how you determined it?visually? 

What reported in subsection 3.5 it is not correlation but regression. They are not interchangeable.

Correspondingly

3.5.1 Simple linear regression

3.5.2 Multiple linear regression

In table 3, please, provide the level of significane for each parameter.

lines 268-272 - This is methodical part. It is repeated several times further in the text. So I suggest you put it in Methods section. And describe the validation part there.

line 281-283 please, clarify how exactly CR was included. And this is better to be placed in Methods section.

In table 5, please, provide the level of significane for each parameter.

And also specify not only coefficients but also the parameters (wavebands) included in the models. Because, this is actually important. And can be used further in practice. You should give this part more attention in your study.

line 339 - humidity is usually used for air, for soil it is moisture content.

line 344 - the same type of soil samples - which types you are referring to?

You need to correct Author Contributions and Conflicts of Interests and leave only relevant information. 

Reviewer 2 Report

Revision of the manuscript ID: remotesensing-402157 “Correlation between spectral characteristics and physiochemical parameters of soda-saline soils in different states”.

Comments to the authors

General comments

The article deals with the study of soil surface spectra, with soil samples prepared under lab conditions using three extreme surface conditions, which are mixed under natural conditions. The idea of the article is original and merits publication in RS journal. However, the article is not ready for publication, it needs to be improved in terms of wording. In general the article needs to clarify many sentences, and eliminate vague comments. The use of technical (Statistical, Pedological, etc.) terminology must be appropriate and precise.

Literature about the use of soil cracking ratio is missing.

Specific comments:

L30 International System units.

L31. Please, add an appropriate reference for global data about salinity /salinization. The

L32 We have “a quick and effective method for measuring soil salinity”, and it is electrical conductivity. Maybe you are considering other aspects of the subject. “and protect the ecosystem” is too vague and unclear.

L33 – 34. The Metternicht study is a revision of potentials and constraints, I have doubts about the discussion on internal structures (of what: Surface salts? Soil structure?) in this study.

L36: remove “level”

L39: lab or field experiments? Do you refer to measurements? Please explain a little the experiment (no English text available) as we need information to understand the article. The spectral characteristics of salts were studied by Hunt et al in 1972.

L41 spectral absorption “features”? please clarify.

L43: Explain salinity “information”

L45 ” degree of humiliation of dissolved organic matters increases when the ion exchange concentration increases”. Please correct / clarify what is the spectral information here.

L49-53. I am not sure if the revision about spectra data bases are in line with the objective of this article.

L51 reference 11 should be replaced by Earth-Science Reviews 155 (2016) 198–230

L 55 clay mineralogy?

L61 I do not understand “spectra data points”, do you mean spectra? Or spectral data?

L62 If possible, give the name of the instrument /or not, in all the mentioned works, for consistency.

L66 Explain “soil category” and remove “level”. Field spectra from soil only represent the soil surface (the very upper layer). Soil spectra from grinded soil samples, do not represent the three dimensions of the pedon. You should correct the sentence in this sense.

L67: can be predicted? Modelled? Estimated? But not obtained.

L69 clay-rich soils

L73-74 remove “level” Could you add a reference about this significant statement? Can you add literature revision on the degree of cracking

L74. Can you remove “overall”?

L77. remove “, respectively,” Improve the wording of the last sentence , is repetitive.

L82 Revise wording.

L86 source of climatic data is missing. Clarify which evaporation  (evapotranspiration  or potential evaporation? Replace “huge difference” by the most appropriate “ hydric deficit”. Clarify “special” landscape.

L89. Please, explain the heterogeneity of soil (composition, color, surface features, etc…) and that of environmental factors.

L90 At sites where soil had different salinity degrees?, at which temperature were the samples oven-dried? Salts can be transformed during drying.

L97-98: please, indicate the relation water:soil of the slurry and explain how you smoothed their surfaces.

L102. Could you add camera and objective data, photograph settings, and lightening. Photographs are later used for CR calculation. Since these photographs are used for CR calculation, you could integrate both subsections 2.2 and 2.5 in only one, at least from L102-L110.

L116 and 118. I understand that the symbol used (approx.) should be a hyphen (interval)

L143 Please, explain how did you manage the problem of shadows, which are visible in Figure 2, at the two distances, 15 cm and 1 m.

L145. You should give explanations about the number of measurements per sample (powdered/ cracked /aggregates). For cracked soils, I understand each reading encompass the whole sample 50x50?

L152. Write with whole words CR here

L153 It is not clear to me what the samples with aggregates are (Not specified in section b). And how do you manage to obtain them.

L158. Equation. It is obvious that shadows generated in the soil surface due to cracks condition the reflectance. However, I do not see it is obvious from equation L156.

L16 ” image was cut to keep the cracked areas”. It was understood that the photograph fits exactly the size of the sample, 50x50.

L179. I do not understand the meaning of “characteristic”, if you did a previous selection of bands.

L182. Explain why you selected these parameters to be correlated to the spectra, based on the classic literature about spectral behaviour of different salts (mainly determined in laboratory). I think clay mineralogy (Since clay content is very similar, low CV) and organic matter are a key factors conditioning soil cracking and soil spectra.

L191. Add EC after electrical conductivity in L182

L195 and Table 1. Use only one decimal.

L205. Table 1must read these are data from 57 soil samples. Standard deviation? The methodologies for determinations in Table 1 must be explained previously, in the corresponding section.

L207 to 210. These sentence should be in methodology.

L231-214. Redundant.

L215. Indicate the measure unit of CR. In fact, Table 2 and Figure 4 can be integrated in only one Figure. Otherwise, the basic statistics (mean, etc.) can self-explain the CR values range and variability.

L219. Methodology applied to obtain the soil “smooth aggregates” is missing.

L220. You should compare the reflectance of samples with different salt content.

L222-223. The spectral values in Figure 5 do not match the explanation in the text. Soil aggregate spectra show the lowest values.

L225-230. Minerals are not mentioned (halite, etc.), you indicate salt composition. Specify the mineral components of your soils.

L232. Use spectral curve or reflectance. Of the same soil type? Under different surface states?. I do not understand which “Reflectance is a ratio” means. Please explain.

L235 Pearson correlation, r? Coefficients of determination, R2? Please, specify.

L246 Remove “level”

L247-249. I do not understand this sentence, no statistical significance is shown and correlation values are missing.

L3.5.1 and 3.5.2. Title is complicated to follow, please simplify to clarify. “characteristics of multiple characteristic…”

L263-264. Here you use the coefficient of determination, which must indicate a “certain relationship, or a part of the spectra is explained by this factor”. Revise the use of “correlation” in all the manuscript and adjust terminology. The regression is calculated for how many bands of the continuous spectra?

L299 Before and after must be indicated in Table 4.

L305 to 310. You should indicate the statistical significance. R2 values are not really different.

L337. The spectra or the spectral curves. Remove “level” when using salinity, through all the manuscript.

L338 your own research?

L340. The soil reflectance decreases.

L341 the soil reflectance …decreases with soil salinity.

L242. ·”same humidity” Please be aware that it depends on salt type, since hygroscopic salts may have other behaviour due to water content.

L347. Information about aggregates size is missing.

L349. …influenced by soil mineralogy…

L350 Fabric? Texture: do haw it influences reflectance,? Do you mean clay content (mineralogy and moisture)?

L352-353. “the same clay contents and mineral composition”. The same clay percentage is clear, but up to this moment, the mineral composition is not shown in this study.

L355-357. Please, add a reference for the statement ” the category of no activity soil”, did you mean active soil? Soil with 50% smectites is considered non active, from which point of view?

L358-359. I do not agree with this sentence” Though the clay contents have little effect on the swell-shrink characteristics of soils”. Please explain. This discussion about clay mineralogy and cracking is very interesting. However, you have not shown clay mineralogy data of your 57 soil samples.

L377-378. I agree with the shadows influence on reflectance. You should check literature for discussion. Remove “reflection energy”.

L396. I do not understand “would effectively improve the high spectral inversion accuracy”

L412 Soil reflectance decreases with salinity.

Discussion related to the occurrence of mixing states (not these three pure conditions) under natural conditions, would be important.

Reviewer 3 Report

I went through the manuscript entitled as "Correlation between spectral characteristics and physiochemical parameters of soda-saline soils in different states”. This manuscript deals to build a correlation between spectral and physiochemical properties of saline soils. The manuscript is well designed and the paper is basically correctly written. However, there are some issues both on the research and on its presentation. They are listed below:

Abstract: 

Page 1, Line 18: …..physiochemical parameters (salinity, Na+, pH level, and EC)…… what the salinity means here? Do you mean EC? All of the mentioned parameters are chemical properties? Could you please justify this?

Page 1, lines 24-25: please rewrite these sentence. It is unclear and hard to understand.

Introduction:

General comments:

Page1, Lines 30-31: …..million hm2…. What is “hm2”? please use IS units or introduce it at the first place of use.

Lines 34, 233, 256: “it’s” This is not usually used in formal writing, so use complete form through the MS “it is”.

Page 2, Lines 45: degree of humiliation of dissolved organic matters. Please check “humiliation”, Is it an appropriate word for here? please check.

Page 2, Line 46: Matinfar" et al"

Page 2, Line 51: The Viscarra Rossel group [11]. This unusual form of citing. The readers will assume that “group” is a part of the family name of the investigator, so it is necessary to remove “group”.

Page 2, Lines 60-65: Shepherd et al. [14] obtained ……… Long sentence. Please divide it into independent sentences.

Materials and Methods

It is very good to present the geographical situation of China, Jilin province, and the study area as figure 1.

Page 3, Lines 89-91: the sampling scheme is vague.  How did you choose the sampling locations? Is it cover all diversity in the study area? The number of samples is enough for this large area? Please clear.

Page 4, Line 152 : Calculation of the soil crack ratio

Results

Page6, Line 194: Please rewrite this sentence.

Table 1. It is necessary to mention that what kind of physiochemical parameters measured. Then you can say we used routine methods to measure them.

Page 6, Line 196: ds/m to 3.39 ds/m…. one unit is enough. Please remove the first one.

Page 6, Line 197: The total salinity varied significantly….. please explain how to calculate the “total salinity” in material and methods.

Table 2: please provide table 2 in landscape orientation. It is hard to follow through it. Then you can use the same pattern as samples presented in Figure 4.

Page10, Line 272 : in Abstract section mentioned “mean absolute error (MAE)” used for validation but in this line used “MAE (mean average error)” please keep constant through MS.

Table 4: The header of the table is not readable. “Sa” stand for salinity, if yes, it should be identified at the below of the table.

Conclusions

The conclusions section is more or less like Abstract. There is a lack of a concluding sentence or paragraph about the advantage of the proposed method and the future usage and research.

References

The references should check again. There is some mistake in citing through the MS and also formatting e.g. numbers 7, 17, 19, 21, 22 and ….